# Variational Inference of Disentangled Latent Concepts from Unlabeled Observations

**Abhishek Kumar, Prasanna Sattigeri, Avinash Balakrishnan**
IBM Research AI
Yorktown Heights, NY
{abhishk,psattig,avinash.bala}@us.ibm.com

## Abstract

Disentangled representations, where the higher level data generative factors are reflected in disjoint latent dimensions, offer several benefits such as ease of deriving invariant representations, transferability to other tasks, interpretability, etc. We consider the problem of unsupervised learning of disentangled representations from large pool of unlabeled observations, and propose a variational inference based approach to infer disentangled latent factors. We introduce a regularizer on the expectation of the approximate posterior over observed data that encourages the disentanglement. We also propose a new disentanglement metric which is better aligned with the qualitative disentanglement observed in the decoder's output. We empirically observe significant improvement over existing methods in terms of both disentanglement and data likelihood (reconstruction quality).

## 1 Introduction

Feature representations of the observed raw data play a crucial role in the success of machine learning algorithms. Effective representations should be able to capture the underlying (abstract or high-level) latent generative factors that are relevant for the end task while ignoring the inconsequential or nuisance factors. *Disentangled* feature representations have the property that the generative factors are revealed in disjoint subsets of the feature dimensions, such that a change in a single generative factor causes a highly sparse change in the representation. Disentangled representations offer several advantages – *(i) Invariance:* it is easier to derive representations that are invariant to nuisance factors by simply marginalizing over the corresponding dimensions, *(ii) Transferability:* they are arguably more suitable for transfer learning as most of the key underlying generative factors appear segregated along feature dimensions, *(iii) Interpretability:* a human expert may be able to assign meanings to the dimensions, *(iv) Conditioning and intervention:* they allow for interpretable conditioning and/or intervention over a subset of the latents and observe the effects on other nodes in the graph. Indeed, the importance of learning disentangled representations has been argued in several recent works (Bengio et al., 2013; Lake et al., 2016; Ridgeway, 2016).

Recognizing the significance of disentangled representations, several attempts have been made in this direction in the past (Ridgeway, 2016). Much of the earlier work assumes some sort of supervision in terms of: (i) partial or full access to the generative factors per instance (Reed et al., 2014; Yang et al., 2015; Kulkarni et al., 2015; Karaletsos et al., 2015), (ii) knowledge about the nature of generative factors (e.g, translation, rotation, etc.) (Hinton et al., 2011; Cohen & Welling, 2014), (iii) knowledge about the changes in the generative factors across observations (e.g., sparse changes in consecutive frames of a Video) (Goroshin et al., 2015; Whitney et al., 2016; Fraccaro et al., 2017; Denton & Birodkar, 2017; Hsu et al., 2017), (iv) knowledge of a complementary signal to infer representations that are conditionally independent of it[1] (Cheung et al., 2014; Mathieu et al., 2016; Siddharth et al., 2017). However, in most real scenarios, we only have access to raw observations without any supervision about the generative factors. It is a challenging problem and many of the earlier attempts have not been able to scale well for realistic settings (Schmidhuber, 1992; Desjardins et al., 2012; Cohen & Welling, 2015) (see also, Higgins et al. (2017)).

---

[1] The representation itself can still be entangled in rest of the generative factors.

Recently, Chen et al. (2016) proposed an approach to learn a generative model with disentangled factors based on Generative Adversarial Networks (GAN) (Goodfellow et al., 2014), however implicit generative models like GANs lack an effective inference mechanism[2], which hinders its applicability to the problem of learning disentangled representations. More recently, Higgins et al. (2017) proposed an approach based on Variational AutoEncoder (VAE) Kingma & Welling (2013) for inferring disentangled factors. The inferred latents using their method (termed as $\beta$-VAE ) are empirically shown to have better disentangling properties, however the method deviates from the basic principles of variational inference, creating increased tension between observed data likelihood and disentanglement. This in turn leads to poor quality of generated samples as observed in (Higgins et al., 2017).

In this work, we propose a principled approach for inference of disentangled latent factors based on the popular and scalable framework of amortized variational inference (Kingma & Welling, 2013; Stuhlmüller et al., 2013; Gershman & Goodman, 2014; Rezende et al., 2014) powered by stochastic optimization (Hoffman et al., 2013; Kingma & Welling, 2013; Rezende et al., 2014). Disentanglement is encouraged by introducing a regularizer over the induced *inferred prior*. Unlike $\beta$-VAE (Higgins et al., 2017), our approach does not introduce any extra conflict between disentanglement of the latents and the observed data likelihood, which is reflected in the overall quality of the generated samples that matches the VAE and is much better than $\beta$-VAE. This does *not* come at the cost of higher entanglement and our approach also outperforms $\beta$-VAE in disentangling the latents as measured by various quantitative metrics. We also propose a new disentanglement metric, called *Separated Attribute Predictability* or SAP, which is better aligned with the qualitative disentanglement observed in the decoder's output compared to the existing metrics.

## 2 FORMULATION

We start with a generative model of the observed data that first samples a latent variable $\mathbf{z} \sim p(\mathbf{z})$, and an observation is generated by sampling from $p_\theta(\mathbf{x}|\mathbf{z})$. The joint density of latents and observations is denoted as $p_\theta(\mathbf{x}, \mathbf{z}) = p(\mathbf{z})p_\theta(\mathbf{x}|\mathbf{z})$. The problem of inference is to compute the posterior of the latents conditioned on the observations, i.e., $p_\theta(\mathbf{z}|\mathbf{x}) = \frac{p_\theta(\mathbf{x}, \mathbf{z})}{\int p_\theta(\mathbf{x}, \mathbf{z})d\mathbf{z}}$. We assume that we are given a finite set of samples (observations) from the true data distribution $p(\mathbf{x})$. In most practical scenarios involving high dimensional and complex data, this computation is intractable and calls for approximate inference. Variational inference takes an optimization based approach to this, positing a family $\mathcal{D}$ of approximate densities over the latents and reducing the approximate inference problem to finding a member density that minimizes the Kullback-Leibler divergence to the true posterior, i.e., $q_{\mathbf{x}}^* = \min_{q \in \mathcal{D}} \mathrm{KL}(q(\mathbf{z})\|p_\theta(\mathbf{z}|\mathbf{x}))$ (Blei et al., 2017). The idea of amortized inference (Kingma & Welling, 2013; Stuhlmüller et al., 2013; Gershman & Goodman, 2014; Rezende et al., 2014) is to explicitly share information across inferences made for each observation. One successful way of achieving this for variational inference is to have a so-called *recognition model*, parameterized by $\phi$, that encodes an inverse map from the observations to the approximate posteriors (also referred as variational autoencoder or VAE) (Kingma & Welling, 2013; Rezende et al., 2014). The recognition model parameters are learned by optimizing the problem $\min_\phi \mathbb{E}_{\mathbf{x}}\mathrm{KL}(q_\phi(\mathbf{z}|\mathbf{x})\|p_\theta(\mathbf{z}|\mathbf{x}))$, where the outer expectation is over the true data distribution $p(\mathbf{x})$ which we have samples from. This can be shown as equivalent to maximizing what is termed as evidence lower bound (ELBO):

$$\arg\min_{\theta,\phi} \mathbb{E}_{\mathbf{x}}\mathrm{KL}(q_\phi(\mathbf{z}|\mathbf{x})\|p_\theta(\mathbf{z}|\mathbf{x})) = \arg\max_{\theta,\phi} \mathbb{E}_{\mathbf{x}} \left[ \mathbb{E}_{\mathbf{z}\sim q_\phi(\mathbf{z}|\mathbf{x})} \left[\log p_\theta(\mathbf{x}|\mathbf{z}\right] - \mathrm{KL}(q_\phi(\mathbf{z}|\mathbf{x})\|p(\mathbf{z})) \right]$$

(1)

The ELBO (the objective at the right side of Eq. 1) lower bounds the log-likelihood of observed data, and the gap vanishes at the global optimum. Often, the density forms of $p(\mathbf{z})$ and $q_\phi(\mathbf{z}|\mathbf{x})$ are chosen such that their KL-divergence can be written analytically in a closed-form expression (e.g., $p(\mathbf{z})$ is $N(0, I)$ and $q_\phi(\mathbf{z}|\mathbf{x})$ is $N(\mu_\phi(\mathbf{x}), \Sigma_\phi(\mathbf{x}))$) (Kingma & Welling, 2013). In such cases, the ELBO can be efficiently optimized (to a stationary point) using stochastic first order methods where both expectations are estimated using mini-batches. Further, in cases when $q_\phi(\cdot)$ can be written as a continuous transformation of a fixed base distribution (e.g., the standard normal distribution), a low

---

[2]There have been a few recent attempts in this direction for visual data (Dumoulin et al., 2016; Donahue et al., 2016; Kumar et al., 2017) but often the reconstructed samples are semantically quite far from the input samples, sometimes even changing in the object classes.

variance estimate of the gradient over $\phi$ can be obtained by coordinate transformation (also referred as reparametrization) (Fu, 2006; Kingma & Welling, 2013; Rezende et al., 2014).

## 2.1 GENERATIVE STORY: DISENTANGLED PRIOR

Most VAE based generative models for real datasets (e.g., text, images, etc.) already work with a relatively simple and disentangled prior $p(\mathbf{z})$ having no interaction among the latent dimensions (e.g., the standard Gaussian $N(0, I)$) (Bowman et al., 2015; Miao et al., 2016; Hou et al., 2017; Zhao et al., 2017). The complexity of the observed data is absorbed in the conditional distribution $p_\theta(\mathbf{x}|\mathbf{z})$ which encodes the interactions among the latents. Hence, as far as the generative modeling is concerned, disentangled prior sets us in the right direction.

## 2.2 INFERRING DISENTANGLED LATENTS

Although the generative model starts with a disentangled prior, our main objective is to *infer* disentangled latents which are potentially conducive for various goals mentioned in Sec. 1 (e.g., invariance, transferability, interpretability). To this end, we consider the density over the inferred latents induced by the approximate posterior inference mechanism,

$$q_\phi(\mathbf{z}) = \int q_\phi(\mathbf{z}|\mathbf{x})p(\mathbf{x})d\mathbf{x}, \tag{2}$$

which we will subsequently refer to as the *inferred prior* or *expected variational posterior* ($p(\mathbf{x})$ is the true data distribution that we have only samples from). For inferring disentangled factors, this should be factorizable along the dimensions, i.e., $q_\phi(\mathbf{z}) = \prod_i q_i(z_i)$, or equivalently $q_{i|j}(z_i|z_j) = q_i(z_i), \forall i, j$. This can be achieved by minimizing a suitable distance between the inferred prior $q_\phi(\mathbf{z})$ and the disentangled generative prior $p(\mathbf{z})$. We can also define *expected posterior* as $p_\theta(\mathbf{z}) = \int p_\theta(\mathbf{z}|\mathbf{x})p(\mathbf{x})d\mathbf{x}$. If we take KL-divergence as our choice of distance, by relying on its pairwise convexity (i.e., $\mathrm{KL}(\lambda p_1 + (1-\lambda)p_2 \| \lambda q_1 + (1-\lambda)q_2) \leq \lambda \mathrm{KL}(p_1\|q_1) + (1-\lambda)\mathrm{KL}(p_2\|q_2)$) (Van Erven & Harremos, 2014), we can show that the distance between $q_\phi(\mathbf{z})$ and $p_\theta(\mathbf{z})$ is bounded by the objective of the variational inference:

$$\mathrm{KL}(q_\phi(\mathbf{z})\|p_\theta(\mathbf{z})) = \mathrm{KL}(\mathbb{E}_{\mathbf{x}\sim p(\mathbf{x})}q_\phi(\mathbf{z}|\mathbf{x})\|\mathbb{E}_{\mathbf{x}\sim p(\mathbf{x})}p_\theta(\mathbf{z}|\mathbf{x})) \leq \mathbb{E}_{\mathbf{x}\sim p(\mathbf{x})}\mathrm{KL}(q_\phi(\mathbf{z}|\mathbf{x})\|p_\theta(\mathbf{z}|\mathbf{x})). \tag{3}$$

In general, the prior $p(\mathbf{z})$ and expected posterior $p_\theta(\mathbf{z})$ will be different, although they may be close (they will be same when $p_\theta(\mathbf{x}) = \int p_\theta(\mathbf{x}|\mathbf{z})p(\mathbf{z})d\mathbf{z}$ is equal to $p(\mathbf{x})$). Hence, variational posterior inference of latent variables with disentangled prior naturally encourages inferring factors that are *close* to being disentangled. We think this is the reason that the original VAE (Eq. (1)) has also been observed to exhibit some disentangling behavior on simple datasets such as MNIST (Kingma & Welling, 2013). However, this behavior does not carry over to more complex datasets (Aubry et al., 2014; Liu et al., 2015; Higgins et al., 2017), unless extra supervision on the generative factors is provided (Kulkarni et al., 2015; Karaletsos et al., 2015). This can be due to: (i) $p(\mathbf{x})$ and $p_\theta(\mathbf{x})$ being far apart which in turn causes $p(\mathbf{z})$ and $p_\theta(\mathbf{z})$ being far apart, and (ii) the non-convexity of the ELBO objective which prevents us from achieving the global minimum of $\mathbb{E}_{\mathbf{x}}\mathrm{KL}(q_\phi(\mathbf{z}|\mathbf{x})\|p_\theta(\mathbf{z}|\mathbf{x}))$ (which is 0 and implies $\mathrm{KL}(q_\phi(\mathbf{z})\|p_\theta(\mathbf{z})) = 0$). In other words, maximizing the ELBO (Eq. (1)) might also result in reducing the value of $\mathrm{KL}(q_\phi(\mathbf{z})\|p(\mathbf{z}))$, however, due to the aforementioned reasons, the gap between $\mathrm{KL}(q_\phi(\mathbf{z})\|p(\mathbf{z}))$ and $\mathbb{E}_{\mathbf{x}}\mathrm{KL}(q_\phi(\mathbf{z}|\mathbf{x})\|p_\theta(\mathbf{z}|\mathbf{x}))$ could be large at the stationary point of convergence. Hence, minimizing $\mathrm{KL}(q_\phi(\mathbf{z})\|p(\mathbf{z}))$ or any other suitable distance $D(q_\phi(\mathbf{z}), p(\mathbf{z}))$ explicitly will give us better control on the disentanglement. This motivates us to add $D(q_\phi(\mathbf{z})\|p(\mathbf{z}))$ as part of the objective to encourage disentanglement during inference, i.e.,

$$\max_{\theta,\phi} \mathbb{E}_{\mathbf{x}}\left[\mathbb{E}_{\mathbf{z}\sim q_\phi(\mathbf{z}|\mathbf{x})}\left[\log p_\theta(\mathbf{x}|\mathbf{z})\right] - \mathrm{KL}(q_\phi(\mathbf{z}|\mathbf{x})\|p(\mathbf{z}))\right] - \lambda\, D(q_\phi(\mathbf{z})\|p(\mathbf{z})), \tag{4}$$

where $\lambda$ controls its contribution to the overall objective. We refer to this as DIP-VAE (for Disentangled Inferred Prior) subsequently.

Optimizing (4) directly is not tractable if $D(\cdot, \cdot)$ is taken to be the KL-divergence $\mathrm{KL}(q_\phi(\mathbf{z})\|p(\mathbf{z}))$, which does not have a closed-form expression. One possibility is use the variational formulation of the KL-divergence (Nguyen et al., 2010; Nowozin et al., 2016) that needs only samples from $q_\phi(\mathbf{z})$ and $p(\mathbf{z})$ to estimate a lower bound to $\mathrm{KL}(q_\phi(\mathbf{z})\|p(\mathbf{z}))$. However, this would involve optimizing for

a third set of parameters $\psi$ for the KL-divergence estimator, and would also change the optimization to a saddle-point (min-max) problem which has its own optimization challenges (e.g., gradient vanishing as encountered in training generative adversarial networks with KL or Jensen-Shannon (JS) divergences (Goodfellow et al., 2014; Arjovsky & Bottou, 2017)). Taking $D$ to be another suitable distance between $q_\phi(\mathbf{z})$ and $p(\mathbf{z})$ (e.g., integral probability metrics like Wasserstein distance (Sriperumbudur et al., 2009)) might alleviate some of these issues (Arjovsky et al., 2017) but will still involve complicating the optimization to a saddle point problem in three set of parameters[3]. It should also be noted that using these variational forms of the distances will still leave us with an approximation to the actual distance.

We adopt a simpler yet effective alternative of matching the moments of the two distributions. Matching the covariance of the two distributions will amount to decorrelating the dimensions of $\mathbf{z} \sim q_\phi(\mathbf{z})$ if $p(\mathbf{z})$ is $N(0, I)$. Let us denote $\text{Cov}_{q(\mathbf{z})}[\mathbf{z}] := \mathbb{E}_{q(\mathbf{z})}\left[(\mathbf{z} - \mathbb{E}_{q(\mathbf{z})}[\mathbf{z}])(\mathbf{z} - \mathbb{E}_{q[\mathbf{z}]}(\mathbf{z}))^\top\right]$. By the law of total covariance, the covariance of $\mathbf{z} \sim q_\phi(\mathbf{z})$ is given by

$$\text{Cov}_{q_\phi(\mathbf{z})}[\mathbf{z}] = \mathbb{E}_{p(\mathbf{x})}\text{Cov}_{q_\phi(\mathbf{z}|\mathbf{x})}[\mathbf{z}] + \text{Cov}_{p(\mathbf{x})}\left(\mathbb{E}_{q_\phi(\mathbf{z}|\mathbf{x})}[\mathbf{z}]\right), \qquad (5)$$

where $\mathbb{E}_{q_\phi(\mathbf{z}|\mathbf{x})}[\mathbf{z}]$ and $\text{Cov}_{q_\phi(\mathbf{z}|\mathbf{x})}[\mathbf{z}]$ are random variables that are functions of the random variable $\mathbf{x}$ ($\mathbf{z}$ is marginalized over). Most existing work on the VAE models uses $q_\phi(\mathbf{z}|\mathbf{x})$ having the form $N(\boldsymbol{\mu}_\phi(\mathbf{x}), \boldsymbol{\Sigma}_\phi(\mathbf{x}))$, where $\boldsymbol{\mu}_\phi(\mathbf{x})$ and $\boldsymbol{\Sigma}_\phi(\mathbf{x})$ are the outputs of a deep neural net parameterized by $\phi$. In this case Eq. (5) reduces to $\text{Cov}_{q_\phi(\mathbf{z})}[\mathbf{z}] = \mathbb{E}_{p(\mathbf{x})}[\boldsymbol{\Sigma}_\phi(\mathbf{x})] + \text{Cov}_{p(\mathbf{x})}[\boldsymbol{\mu}_\phi(\mathbf{x})]$, which we want to be close to the Identity matrix. For simplicity, we choose entry-wise squared $\ell_2$-norm as the measure of proximity. Further, $\boldsymbol{\Sigma}_\phi(\mathbf{x})$ is commonly taken to be a diagonal matrix which means that cross-correlations (off-diagonals) between the latents are due to only $\text{Cov}_{p(\mathbf{x})}[\boldsymbol{\mu}_\phi(\mathbf{x})]$. This suggests two possible options for the disentangling regularizer: (i) regularizing only $\text{Cov}_{p(\mathbf{x})}[\boldsymbol{\mu}_\phi(\mathbf{x})]$ which we refer as DIP-VAE-I, (ii) regularizing $\text{Cov}_{q_\phi(\mathbf{z})}[\mathbf{z}]$ which we refer as DIP-VAE-II. Penalizing just the off-diagonals in both cases will lead to lowering the diagonal entries of $\text{Cov}_{p(\mathbf{x})}[\boldsymbol{\mu}_\phi(\mathbf{x})]$ as the $ij$'th off-diagonal is really a derived attribute obtained by multiplying the square-roots of $i$'th and $j$'th diagonals (for each example $\mathbf{x} \sim p(\mathbf{x})$, followed by averaging over all examples). This can be compensated in DIP-VAE-I by a regularizer on the diagonal entries of $\text{Cov}_{p(\mathbf{x})}[\boldsymbol{\mu}_\phi(\mathbf{x})]$ which pulls these towards 1. We opt for two separate hyperparameters controlling the relative importance of the loss on the diagonal and off-diagonal entries as follows:

$$\max_{\theta,\phi} \text{ELBO}(\theta, \phi) - \lambda_{od} \sum_{i \neq j} \left[\text{Cov}_{p(\mathbf{x})}[\boldsymbol{\mu}_\phi(\mathbf{x})]\right]_{ij}^2 - \lambda_d \sum_i \left(\left[\text{Cov}_{p(\mathbf{x})}[\boldsymbol{\mu}_\phi(\mathbf{x})]\right]_{ii} - 1\right)^2. \qquad (6)$$

The regularization terms involving $\text{Cov}_{p(\mathbf{x})}[\boldsymbol{\mu}_\phi(\mathbf{x})]$ in the above objective (6) can be efficiently optimized using SGD, where $\text{Cov}_{p(\mathbf{x})}[\boldsymbol{\mu}_\phi(\mathbf{x})]$ can be estimated using the current minibatch[4].

For DIP-VAE-II, we have the following optimization problem:

$$\max_{\theta,\phi} \text{ELBO}(\theta, \phi) - \lambda_{od} \sum_{i \neq j} \left[\text{Cov}_{q_\phi(\mathbf{z})}[\mathbf{z}]\right]_{ij}^2 - \lambda_d \sum_i \left(\left[\text{Cov}_{q_\phi(\mathbf{z})}[\mathbf{z}]\right]_{ii} - 1\right)^2. \qquad (7)$$

As discussed earlier, the term $\mathbb{E}_{p(\mathbf{x})}\text{Cov}_{q_\phi(\mathbf{z}|\mathbf{x})}[\mathbf{z}]$ contributes only to the diagonals of $\text{Cov}_{q_\phi(\mathbf{z})}[\mathbf{z}]$. Penalizing the off-diagonals of $\text{Cov}_{p(\mathbf{x})}[\boldsymbol{\mu}_\phi(\mathbf{x})]$ in the Objective (7) will contribute to reduction in the magnitude of its diagonals as discussed earlier. As the regularizer on the diagonals is not directly on $\text{Cov}_{p(\mathbf{x})}[\boldsymbol{\mu}_\phi(\mathbf{x})]$, unlike DIP-VAE-I, it will be not be able to keep $[\text{Cov}_{p(\mathbf{x})}[\boldsymbol{\mu}_\phi(\mathbf{x})]]_{ii}$ close to 1: the reduction in $[\text{Cov}_{p(\mathbf{x})}[\boldsymbol{\mu}_\phi(\mathbf{x})]]_{ii}$ will be accompanied by increase in $[\mathbb{E}_{p(\mathbf{x})}\boldsymbol{\Sigma}_\phi(\mathbf{x})]_{ii}$ such that their sum remains close to 1. In datasets where the number of generative factors is less than the latent dimension, DIP-VAE-II is more suitable than DIP-VAE-I as keeping all dimensions *active* might result in *splitting* of an attribute across multiple dimensions, hurting the goal of disentanglement.

It is also possible to match higher order central moments of $q_\phi(\mathbf{z})$ and the prior $p(\mathbf{z})$. In particular, third order central moments (and moments) of the zero mean Gaussian prior are zero, hence $\ell_2$ norm of third order central moments of $q_\phi(\mathbf{z})$ can be penalized.

---

[3]Nonparametric distances like maximum mean discrepancy (MMD) with a characteristic kernel (Gretton et al., 2012) is also an option, however it has its own challenges when combined with stochastic optimization (Dziugaite et al., 2015; Li et al., 2015).

[4]We also tried an alternative of maintaining a running estimate of $\text{Cov}_{p(\mathbf{x})}[\boldsymbol{\mu}_\phi(\mathbf{x})]$ which is updated with every minibatch of $\mathbf{x} \sim p(\mathbf{x})$, however we did not observe a significant improvement over the simpler approach of estimating these using only current minibatch.

## 2.3 COMPARISON WITH $\beta$-VAE

Recently proposed $\beta$-VAE (Higgins et al., 2017) proposes to modify the ELBO by upweighting the $\text{KL}(q_\phi(\mathbf{z}|\mathbf{x})\|p(\mathbf{z}))$ term in order to encourage the inference of disentangled factors:

$$\max_{\theta,\phi} \mathbb{E}_\mathbf{x} \left[ \mathbb{E}_{\mathbf{z}\sim q_\phi(\mathbf{z}|\mathbf{x})} \left[ \log p_\theta(\mathbf{x}|\mathbf{z}) \right] - \beta \, \text{KL}(q_\phi(\mathbf{z}|\mathbf{x})\|p(\mathbf{z})) \right], \tag{8}$$

where $\beta$ is taken to be great than 1. Higher $\beta$ is argued to encourage disentanglement at the cost of reconstruction error (the likelihood term in the ELBO). Authors report empirical results with $\beta$ ranging from 4 to 250 depending on the dataset. As already mentioned, most VAE models proposed in the literature, including $\beta$-VAE, work with $N(\mathbf{0}, \mathbf{I})$ as the prior $p(\mathbf{z})$ and $N(\boldsymbol{\mu}_\phi(\mathbf{x}), \boldsymbol{\Sigma}_\phi(\mathbf{x}))$ with diagonal $\boldsymbol{\Sigma}_\phi(\mathbf{x})$ as the approximate posterior $q_\phi(\mathbf{z}|\mathbf{x})$. This reduces the objective (8) to

$$\max_{\theta,\phi} \mathbb{E}_\mathbf{x} \left[ \mathbb{E}_{\mathbf{z}\sim q_\phi(\mathbf{z}|\mathbf{x})} \left[ \log p_\theta(\mathbf{x}|\mathbf{z}) \right] - \frac{\beta}{2} \left( \sum_i \left( [\boldsymbol{\Sigma}_\phi(\mathbf{x})]_{ii} - \ln [\boldsymbol{\Sigma}_\phi(\mathbf{x})]_{ii} \right) + \|\boldsymbol{\mu}_\phi(\mathbf{x})\|_2^2 \right) \right]. \tag{9}$$

For high values of $\beta$, $\beta$-VAE would try to pull $\boldsymbol{\mu}_\phi(\mathbf{x})$ towards zero and $\boldsymbol{\Sigma}_\phi(\mathbf{x})$ towards the identity matrix (as the minimum of $x - \ln x$ for $x > 0$ is at $x = 1$), thus making the approximate posterior $q_\phi(\mathbf{z}|\mathbf{x})$ insensitive to the observations. This is also reflected in the quality of the reconstructed samples which is worse than VAE ($\beta = 1$), particularly for high values of $\beta$. Our proposed method does not have such increased tension between the likelihood term and the disentanglement objective, and the sample quality with our method is on par with the VAE.

Finally, we note that both $\beta$-VAE and our proposed method encourage disentanglement of inferred factors by pulling $\text{Cov}_{q_\phi(\mathbf{z})}(\mathbf{z})$ in Eq. (5) towards the identity matrix: $\beta$-VAE attempts to do it by making $\text{Cov}_{q_\phi(\mathbf{z}|\mathbf{x})}(\mathbf{z})$ close to $\mathbf{I}$ and $\mathbb{E}_{q_\phi(\mathbf{z}|\mathbf{x})}(\mathbf{z})$ close to $\mathbf{0}$ individually for all observations $\mathbf{x}$, while the proposed method directly works on $\text{Cov}_{q_\phi(\mathbf{z})}(\mathbf{z})$ (marginalizing over the observations $\mathbf{x}$) which retains the sensitivity of $q_\phi(\mathbf{z}|\mathbf{x})$ to the conditioned-upon observation.

## 3 QUANTIFYING DISENTANGLEMENT: SAP SCORE

Higgins et al. (2017) propose a metric to evaluate the disentanglement performance of the inference mechanism, assuming that the ground truth generative factors are available. It works by first sampling a generative factor $y$, followed by sampling $L$ pairs of examples such that for each pair, the sampled generative factor takes the same value. Given the inferred $\mathbf{z}_x := \boldsymbol{\mu}_\phi(\mathbf{x})$ for each example $\mathbf{x}$, they compute the absolute difference of these vectors for each pair, followed by averaging these difference vectors. This average difference vector is assigned the label of $y$. By sampling $n$ such minibatches of $L$ pairs, we get $n$ such averaged difference vectors for the factor $y$. This process is repeated for all generative factors. A low capacity multiclass classifier is then trained on these vectors to predict the identities of the corresponding generative factors. Accuracy of this classifier on the difference vectors for test set is taken to be a measure of disentanglement. We evaluate the proposed method on this metric and refer to this as **Z-diff score** subsequently.

We observe in our experiments that the Z-diff score (Higgins et al., 2017) is not correlated well with the qualitative disentanglement at the decoder's output as seen in the latent traversal plots (obtained by varying only one latent while keeping the other latents fixed). It also depends on the multiclass classifier used to obtain the score. We propose a new metric, referred as **Separated Attribute Predictability (SAP) score**, that is better aligned with the qualitative disentanglement observed in the latent traversals and also does not involve training any classifier. It is computed as follows: **(i)** We first construct a $d \times k$ score matrix $S$ (for $d$ latents and $k$ generative factors) whose $ij$'th entry is the linear regression or classification score (depending on the generative factor type) of predicting $j$'th factor using only $i$'th latent $[\boldsymbol{\mu}_\phi(\mathbf{x})]_i$. For regression, we take this to be the $R^2$ score obtained with fitting a line (slope and intercept) that minimizes the linear regression error (for the test examples). The $R^2$ score is given by $\left( \frac{\text{Cov}([\boldsymbol{\mu}_\phi(\mathbf{x})]_i, \mathbf{y}_j)}{\sigma_{[\boldsymbol{\mu}_\phi(\mathbf{x})]_i} \sigma_{\mathbf{y}_j}} \right)^2$ and ranges from 0 to 1, with a score of 1 indicating that a linear function of the $i$'th inferred latent explains all variability in the $j$'th generative factor. For classification, we fit one or more thresholds (real numbers) directly on $i$'th inferred latents for the test examples that minimize the balanced classification errors, and take $S_{ij}$ to be the balanced classification accuracy of the $j$'th generative factor. For inactive latent

Table 1: Z-diff score Higgins et al. (2017), the proposed SAP score and reconstruction error (per pixel) on the test sets for 2D Shapes and CelebA ($\beta_1 = 4, \beta_2 = 60, \lambda = 10, \lambda_1 = 5, \lambda_2 = 500$ for 2D Shapes; $\beta_1 = 4, \beta_2 = 32, \lambda = 2, \lambda_1 = 1, \lambda_2 = 80$ for CelebA). For the results on a wider range of hyperparameter values, refer to Fig. 1 and Fig. 2.

| Method | 2D Shapes | | | CelebA | | |
|---|---|---|---|---|---|---|
| | Z-diff | SAP | Reconst. error | Z-diff | SAP | Reconst. error |
| VAE | 81.3 | 0.0417 | 0.0017 | 7.5 | 0.35 | 0.0876 |
| $\beta$-VAE ($\beta{=}\beta_1$) | 80.7 | 0.0811 | 0.0032 | 8.1 | 0.48 | 0.0937 |
| $\beta$-VAE ($\beta{=}\beta_2$) | 95.7 | 0.5503 | 0.0113 | 6.4 | 3.72 | 0.1572 |
| DIP-VAE-I ($\lambda_{od} = \lambda$) | 98.7 | 0.1889 | 0.0018 | 14.8 | 3.69 | 0.0904 |
| DIP-VAE-II ($\lambda_{od} = \lambda_1$) | 95.3 | 0.2188 | 0.0023 | 7.1 | 2.94 | 0.0884 |
| DIP-VAE-II ($\lambda_{od} = \lambda_2$) | 98.0 | 0.5253 | 0.0079 | 11.5 | 3.93 | 0.1477 |

dimensions (having $\sigma_{[\boldsymbol{\mu}_\phi(\mathbf{x})]_i} = [\mathrm{Cov}_{p(x)}[\boldsymbol{\mu}_\phi(\mathbf{x})]]_{ii}$ close to 0), we take $S_{ij}$ to be 0. **(ii)** For each column of the score matrix $S$ which corresponds to a generative factor, we take the difference of top two entries (corresponding to top two most predictive latent dimensions), and then take the mean of these differences as the final **SAP score**. Considering just the top scoring latent dimension for each generative factor is not enough as it does not rule out the possibility of the factor being captured by other latents. A high SAP score indicates that each generative factor is primarily captured in only one latent dimension. Note that a high SAP score does not rule out one latent dimension capturing two or more generative factors well, however in many cases this would be due to the generative factors themselves being correlated with each other, which can be verified empirically using ground truth values of the generative factors (when available). Further, a low SAP score does not rule out good disentanglement in cases when two (or more) latent dimensions might be correlated strongly with the same generative factor and poorly with other generative factors. The generated examples using single latent traversals may not be realistic for such models, and DIP-VAE discourages this from happening by enforcing decorrelation of the latents. However, the SAP score computation can be adapted to such cases by grouping the latent dimensions based on correlations and getting the score matrix at group level, which can be fed as input to the second step to get the final SAP score.

## 4 EXPERIMENTS

We evaluate our proposed method, DIP-VAE, on three datasets – (i) **CelebA** (Liu et al., 2015): It consists of $202,599$ RGB face images of celebrities. We use $64 \times 64 \times 3$ cropped images as used in several earlier works, using $90\%$ for training and $10\%$ for test. (ii) **3D Chairs** (Aubry et al., 2014): It consists of 1393 chair CAD models, with each model rendered from 31 azimuth angles and 2 elevation angles. Following earlier work (Yang et al., 2015; Dosovitskiy et al., 2015) that ignores near-duplicates, we use a subset of 809 chair models in our experiments. We use the binary masks of the chairs as the observed data in our experiments following (Higgins et al., 2017). First $80\%$ of the models are used for training and the rest are used for test. **(iii) 2D Shapes** (Matthey et al., 2017): This is a synthetic dataset of binary 2D shapes generated from the Cartesian product of the shape (heart, oval and square), $x$-position (32 values), $y$-position (32 values), scale (6 values) and rotation (40 values). We consider two baselines for the task of unsupervised inference of disentangled factors: (i) VAE (Kingma & Welling, 2013; Rezende et al., 2014), and (ii) the recently proposed $\beta$-VAE (Higgins et al., 2017). To be consistent with the evaluations in (Higgins et al., 2017), we use the same CNN network architectures (for our encoder and decoder), and same latent dimensions as used in (Higgins et al., 2017) for CelebA, 3D Chairs, 2D Shapes datasets.

**Hyperparameters.** For the proposed DIP-VAE-I, in all our experiments we vary $\lambda_{od}$ in the set $\{1, 2, 5, 10, 20, 50, 100, 500\}$ while fixing $\lambda_d = 10\lambda_{od}$ for 2D Shapes and 3D Chairs, and $\lambda_d = 50\lambda_{od}$ for CelebA. For DIP-VAE-II, we fix $\lambda_{od} = \lambda_d$ for 2D Shapes, and $\lambda_{od} = 2\lambda_d$ for CelebA. Additionally, for DIP-VAE-II we also penalize the $\ell_2$-norm of third order central moments of $q_\phi(\mathbf{z})$ with hyperparameter $\lambda_3 = 200$ for 2D Shapes data ($\lambda_3 = 0$ for CelebA). For $\beta$-VAE, we experiment with $\beta = \{1, 2, 4, 8, 16, 25, 32, 64, 100, 128, 200, 256\}$ (where $\beta = 1$ corresponds to the VAE). We used a batch size of 400 for all 2D Shapes experiments and 100 for all CelebA experiments. For both CelebA and 2D Shapes, we show the results in terms of the Z-diff score Higgins et al. (2017), the

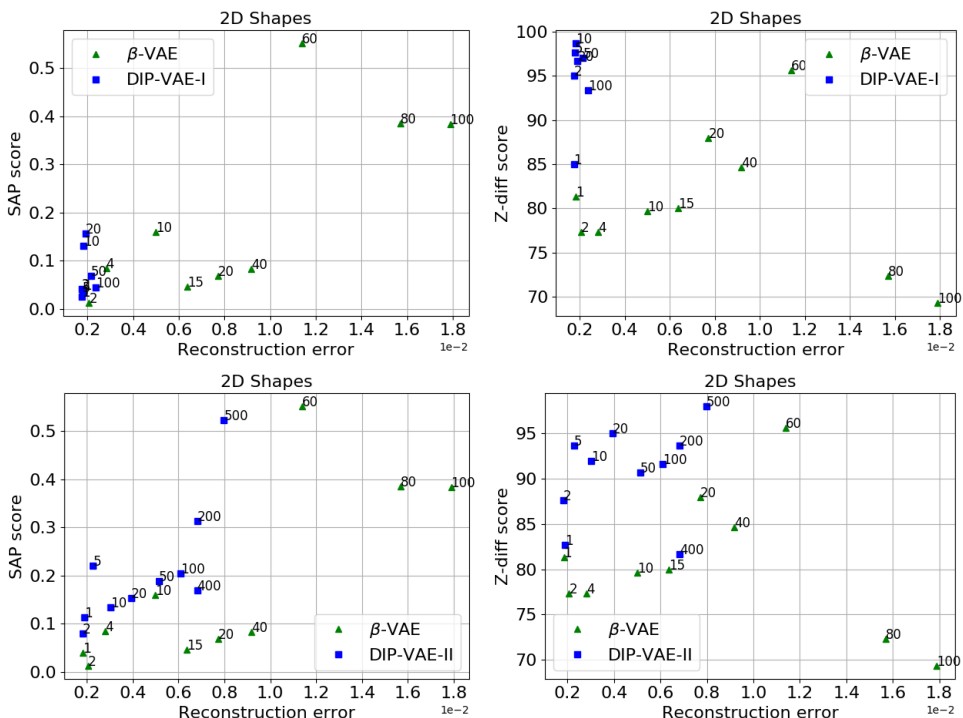

Figure 1: Proposed Separated Atomic Predictability (SAP) score and the Z-diff disentanglement score (Higgins et al., 2017) as a function of average reconstruction error (per pixel) on the test set of 2D Shapes data for $\beta$-VAE and the proposed DIP-VAE. The plots are generated by varying $\beta$ for $\beta$-VAE, and $\lambda_{od}$ for DIP-VAE-I and DIP-VAE-II (the number next to each point is the value of these hyperparameters, respectively).

Table 2: Attribute classification accuracy on CelebA: A classifier $\mathbf{w}^k = \frac{1}{|\mathbf{x}_i:y_i^k=1|} \sum_{\mathbf{x}_i:y_i^k=1} \mu_\phi(\mathbf{x}_i) - \frac{1}{|\mathbf{x}_i:y_i^k=0|} \sum_{\mathbf{x}_i:y_i^k=0} \mu_\phi(\mathbf{x}_i)$ is computed for every attribute $k$ using the training set and a bias is learned by minimizing the hinge loss. Accuracy on other attributes stays about same across all methods.

| Method | Arched Eyebrows | Attractive | Bangs | Black hair | Blond hair | Heavy makeup | Male | Mouth slighly open | No Beard | Wavy hair | Wearing hat | Wearing lipstick |
|---|---|---|---|---|---|---|---|---|---|---|---|---|
| VAE | 71.8 | 73.0 | 89.8 | 78.0 | 88.9 | 79.6 | 83.9 | **76.3** | **87.3** | 70.2 | 95.8 | 83.0 |
| $\beta=2$ | 71.6 | 72.6 | 90.6 | 79.3 | 89.1 | 79.3 | 83.5 | 76.1 | 86.9 | 67.8 | 95.9 | 82.4 |
| $\beta=4$ | 71.6 | 72.6 | 90.0 | 76.6 | 88.9 | 77.8 | 82.3 | 75.7 | 85.3 | 66.8 | 95.8 | 80.6 |
| $\beta=8$ | 71.6 | 71.7 | 90.0 | 76.0 | 87.2 | 76.2 | 80.5 | 73.1 | 85.3 | 63.7 | 95.8 | 79.6 |
| DIP-VAE-I | **73.7** | **73.2** | **90.9** | **80.6** | **91.9** | **81.5** | **85.9** | 75.9 | 85.3 | **71.5** | **96.2** | **84.7** |

proposed SAP score, and reconstruction error. For 3D Chairs data, only two ground truth generative factors are available and the quantitative scores for these are saturated near the peak values, hence we show only the latent traversal plots which we based on our subjective evaluation of the reconstruction quality and disentanglement (shown in Appendix).

**Disentanglement scores and reconstruction error.** For the Z-diff score (Higgins et al., 2017), in all our experiments we use a one-vs-rest linear SVM with weight on the hinge loss $C$ set to 0.01 and weight on the regularizer set to 1. Table 1 shows the Z-diff scores and the proposed SAP

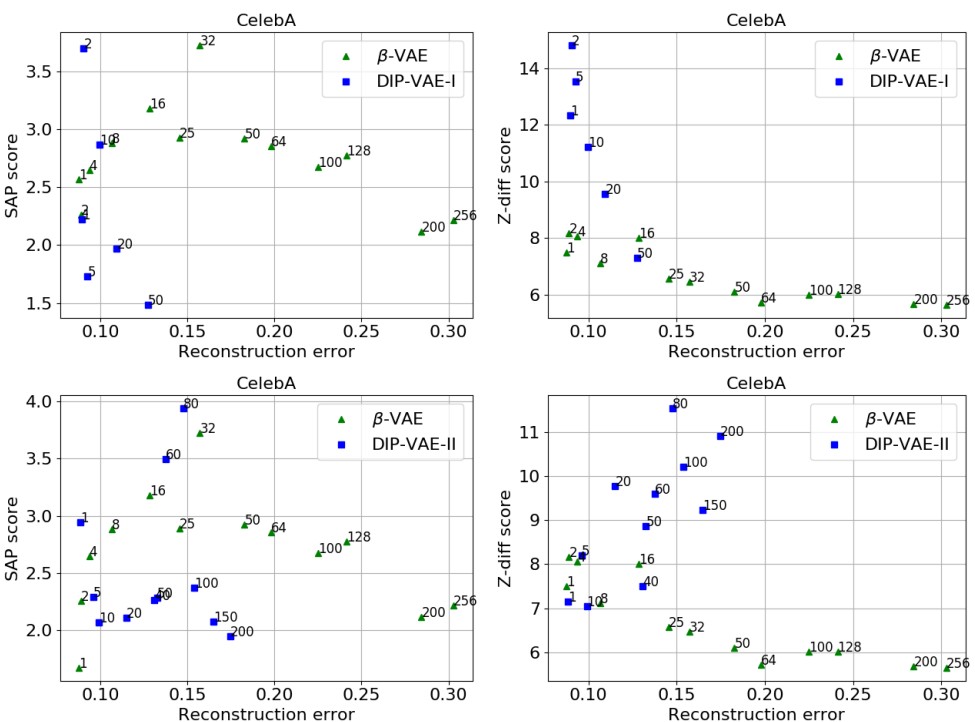

Figure 2: The proposed SAP score and the Z-diff score (Higgins et al., 2017) as a function of average reconstruction error (per pixel) on the test set of CelebA data for $\beta$-VAE and the proposed DIP-VAE. The plots are generated by varying $\beta$ for $\beta$-VAE, and $\lambda_{od}$ for DIP-VAE-I and DIP-VAE-II (the number next to each point is the value of these hyperparameters, respectively).

scores along with reconstruction error (which directly corresponds to the data likelihood) for the test sets of CelebA and 2D Shapes data. Further we also show the plots of how the Z-diff score and the proposed SAP score change with the reconstruction error as we vary the hyperparameter for both methods ($\beta$ and $\lambda_{od}$, respectively) in Fig. 1 (for 2D Shapes data) and Fig. 2 (for CelebA data). The proposed DIP-VAE-I gives much higher Z-diff score at little to no cost on the reconstruction error when compared with VAE ($\beta = 1$) and $\beta$-VAE, for both 2D Shapes and CelebA datasets. However, we observe in the decoder's output for single latent traversals (varying a single latent while keeping others fixed, shown in Fig. 3 and Fig. 4) that a high Z-diff score is not necessarily a good indicator of disentanglement. Indeed, for 2D Shapes data, DIP-VAE-I has a higher Z-diff score (98.7) and almost an order of magnitude lower reconstruction error than $\beta$-VAE for $\beta = 60$, however comparing the latent traversals of $\beta$-VAE in Fig. 3 and DIP-VAE-I in Fig. 4 indicate a better disentanglement for $\beta$-VAE for $\beta = 60$ (though at the cost of much worse reconstruction where every generated sample looks like a hazy blob). On the other hand, we find the proposed SAP score to be correlated well with the qualitative disentanglement seen in the latent traversal plots. This is reflected in the higher SAP score of $\beta$-VAE for $\beta = 60$ than DIP-VAE-I. We also observe that for 2D Shapes data, DIP-VAE-II gives a much better trade-off between disentanglement (measured by the SAP score) and reconstruction error than both DIP-VAE-I and $\beta$-VAE, as shown quantitatively in Fig. 1 and qualitatively in the latent traversal plots in Fig. 3. The reason is that DIP-VAE-I enforces $[\text{Cov}_{p(x)}[\boldsymbol{\mu}_\phi(\mathbf{x})]]_{ii}$ to be close to 1 and this may affect the disentanglement adversely by splitting a generative factor across multiple latents for 2D Shapes where the generative factors are much less than the latent dimension. For real datasets having lots of factors with complex generative processes, such as CelebA, DIP-VAE-I is expected to work well which can be seen in Fig. 2 where DIP-AVE-I yields a much lower reconstruction error with a higher SAP score (as well as higher Z-diff scores).

**Binary attribute classification for CelebA.** We also experiment with predicting the binary attribute values for each test example in CelebA from the inferred $\boldsymbol{\mu}_\phi(\mathbf{x})$. For each attribute $k$, we compute the *attribute vector* $\mathbf{w}^k = \frac{1}{|\mathbf{x}_i:y_i^k=1|} \sum_{\mathbf{x}_i:y_i^k=1} \mu_\phi(\mathbf{x}_i) - \frac{1}{|\mathbf{x}_i:y_i^k=0|} \sum_{\mathbf{x}_i:y_i^k=0} \mu_\phi(\mathbf{x}_i)$ from the training

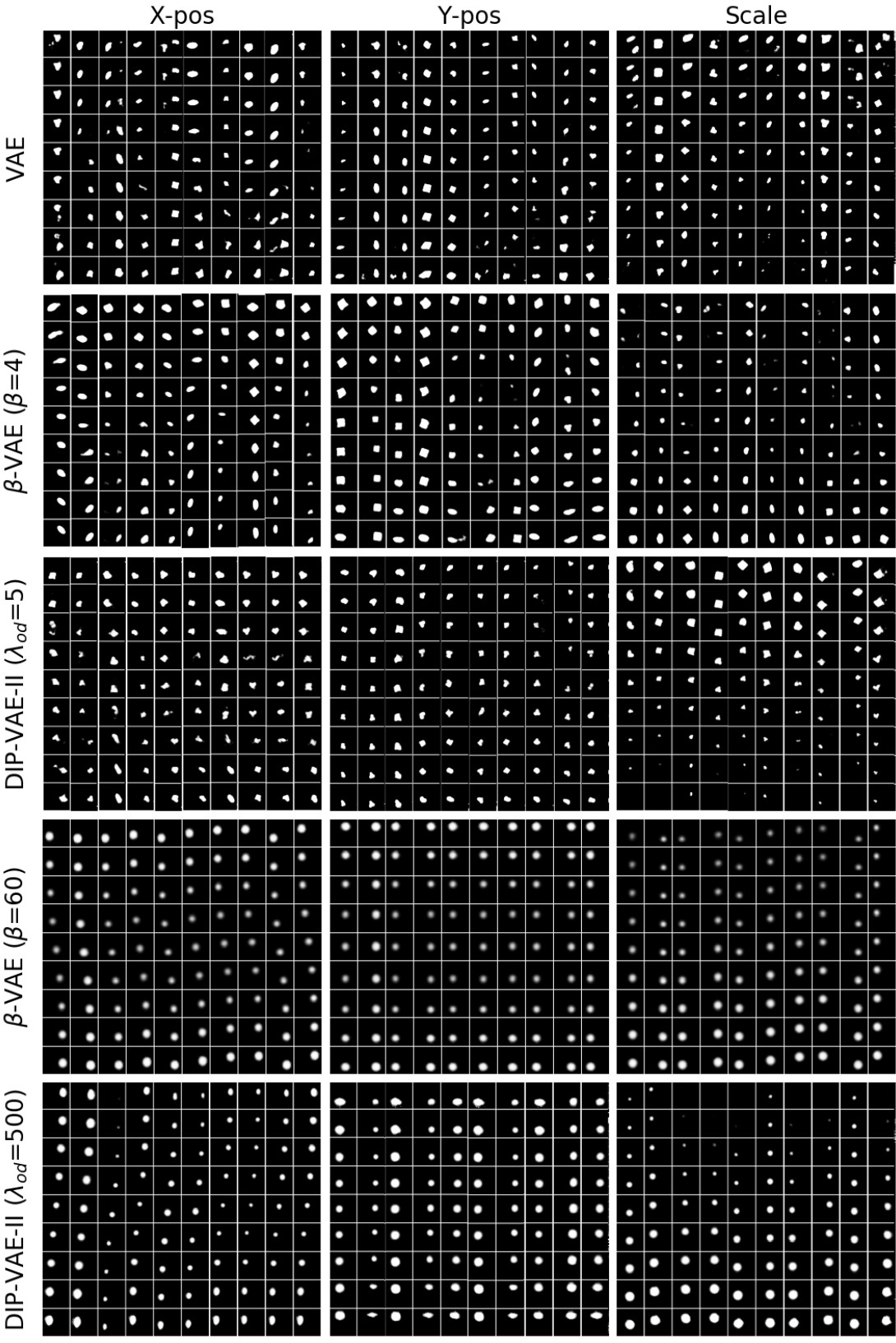

Figure 3: Qualitative results for disentanglement in 2D Shapes dataset (Matthey et al., 2017). SAP scores, Z-diff scores and reconstruction errors for the methods (rows) can be read from Fig. 1.

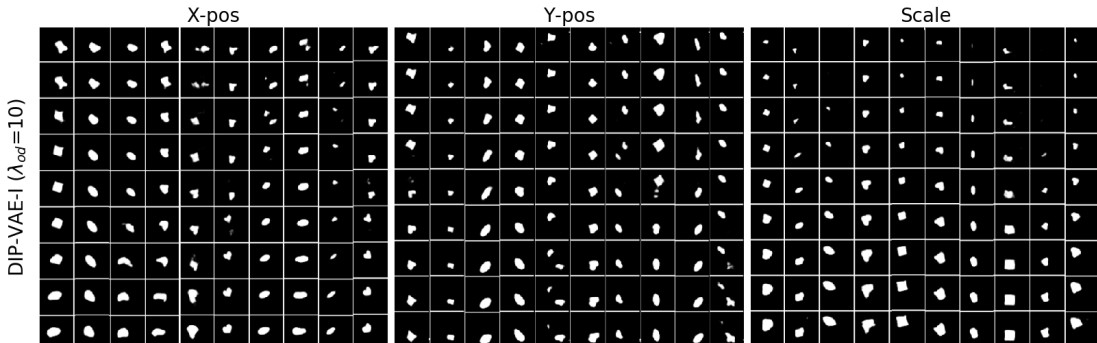

Figure 4: Qualitative results for disentanglement in 2D Shapes dataset (Matthey et al., 2017) for DIP-VAE-I (SAP score 0.1889).

set, and project the $\boldsymbol{\mu}_\phi(\mathbf{x})$ along these vectors. A bias is learned on these scalars (by minimizing hinge loss) which is then used for classifying the test examples. Table 2 shows the results for the attribute which show the highest change across various methods (most other attribute accuracies do not change). The proposed DIP-VAE outperforms both VAE and $\beta$-VAE for most attributes. The performance of $\beta$-VAE gets worse as $\beta$ is increased further.

## 5 RELATED WORK

Adversarial autoencoder (Makhzani et al., 2015) also matches $q_\phi(z)$ (which is referred as *aggregated posterior* in their work) to the prior $p(z)$. However, adversarial autoencoder does not have the goal of minimizing $\text{KL}(q_\phi(\mathbf{z}|\mathbf{x})||p_\theta(\mathbf{z}|\mathbf{x}))$ which is the primary goal of variational inference. It maximizes $\mathbb{E}_\mathbf{x}\left[\mathbb{E}_{\mathbf{z}\sim q_\phi(\mathbf{z}|\mathbf{x})}\left[\log p_\theta(\mathbf{x}|\mathbf{z})\right]\right] - \lambda\, D(q_\phi(\mathbf{z})||p(\mathbf{z}))$, where $D$ is the distance induced by a *discriminator* that tries to classify $\mathbf{z}\sim q_\phi(\mathbf{z})$ from $\mathbf{z}\sim p(\mathbf{z})$ by optimizing a cross-entropy loss (which induces JS-divergence as $D$). This can be contrasted with the objective in (4).

**Invariance and Equivariance.** Disentanglement is closely connected to invariance and equivariance of representations. If $R : \mathbf{x} \to \mathbf{z}$ is a function that maps the observations to the feature representations, equivariance (with respect to $T$) implies that a primitive transformation $T$ of the input results in a corresponding transformation $T'$ of the feature, i.e., $R(T(\mathbf{x})) = T'(R(\mathbf{x}))$. Disentanglement requires that $T'$ acts only on a small subset of dimensions of $R(\mathbf{x})$ (a sparse action). In this sense, equivariance is a more general notion encompassing disentanglement as a special case, however this special case carries additional benefits of interpretability, ease of transferrability, etc. Invariance is also a special case of equivariance which requires $T'$ to be identity for $R$ to be invariant to the action of $T$ on the input observations. However, invariance can obtained more easily from disentangled representations than from equivariant representations by simply marginalizing the appropriate subset of dimensions. There exists a lot of prior work in the literature on equivariant and invariant feature learning, mostly under the supervised setting which assumes the knowledge about the nature of input transformations (e.g., rotations, translations, scaling for images, etc.) (Schmidt & Roth, 2012; Bruna & Mallat, 2013; Anselmi et al., 2014; 2016; Cohen & Welling, 2016; Dieleman et al., 2016; Haasdonk et al., 2005; Mroueh et al., 2015; Raj et al., 2017).

## 6 CONCLUDING REMARKS

We proposed a principled variational framework to infer disentangled latents from unlabeled observations. Unlike $\beta$-VAE, our variational objective does not have any conflict between the data log-likelihood and the disentanglement of the inferred latents, which is reflected in the empirical results. We also proposed the SAP disentanglement metric that is much better correlated with the qualitative disentanglement seen in the latent traversals than the Z-diff score Higgins et al. (2017). An interesting direction for future work is to take into account the sampling biases in the generative process, both natural (e.g., sampling the *female* gender makes it unlikely to sample *beard* for face images in CelebA) as well as artificial (e.g., a collection of face images that contain much more

smiling faces for males than females misleading us to believe $p(\text{gender,smile}) \neq p(\text{gender})p(\text{smile}))$, which makes the problem challenging and also somewhat less well defined (at least in the case of natural biases). Effective use of disentangled representations for transfer learning is another interesting direction for future work.

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

# Appendix

## A  Latent traversals for 2D Shapes and Chairs dataset

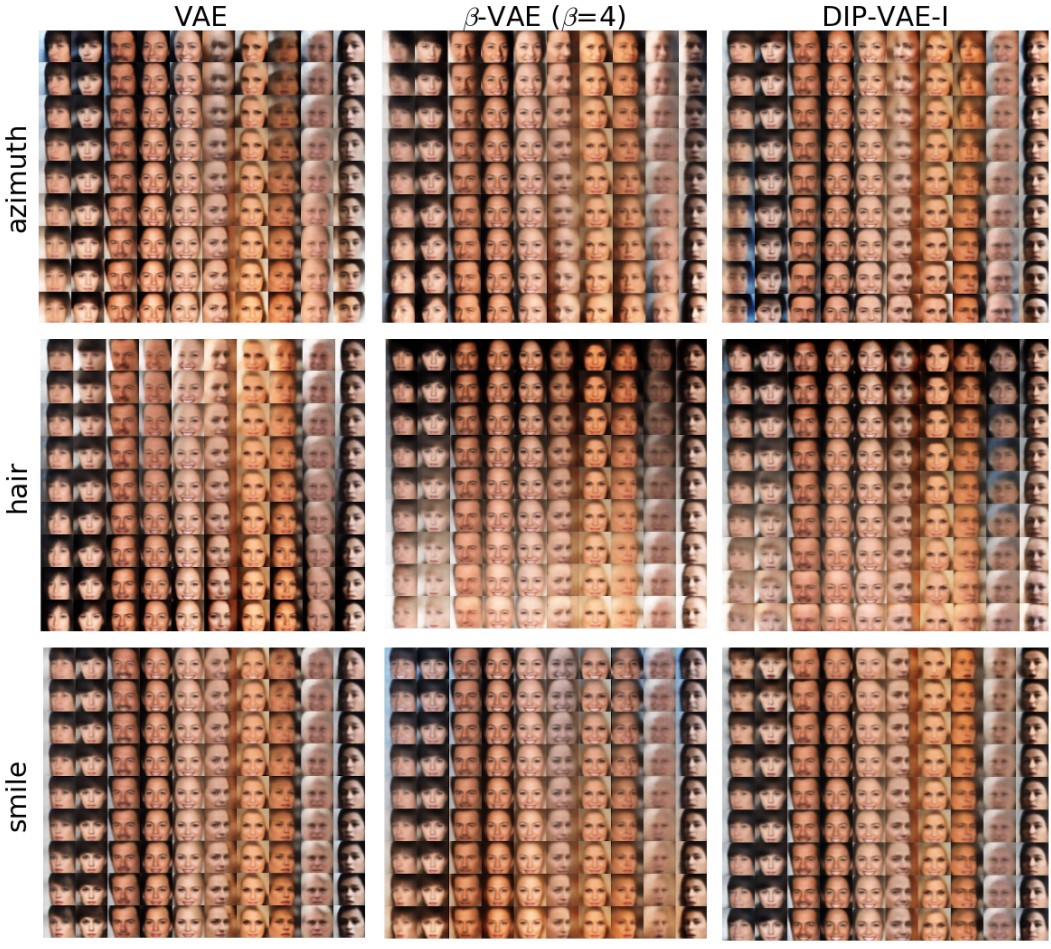

Figure 5: Qualitative results for disentanglement in CelebA dataset.

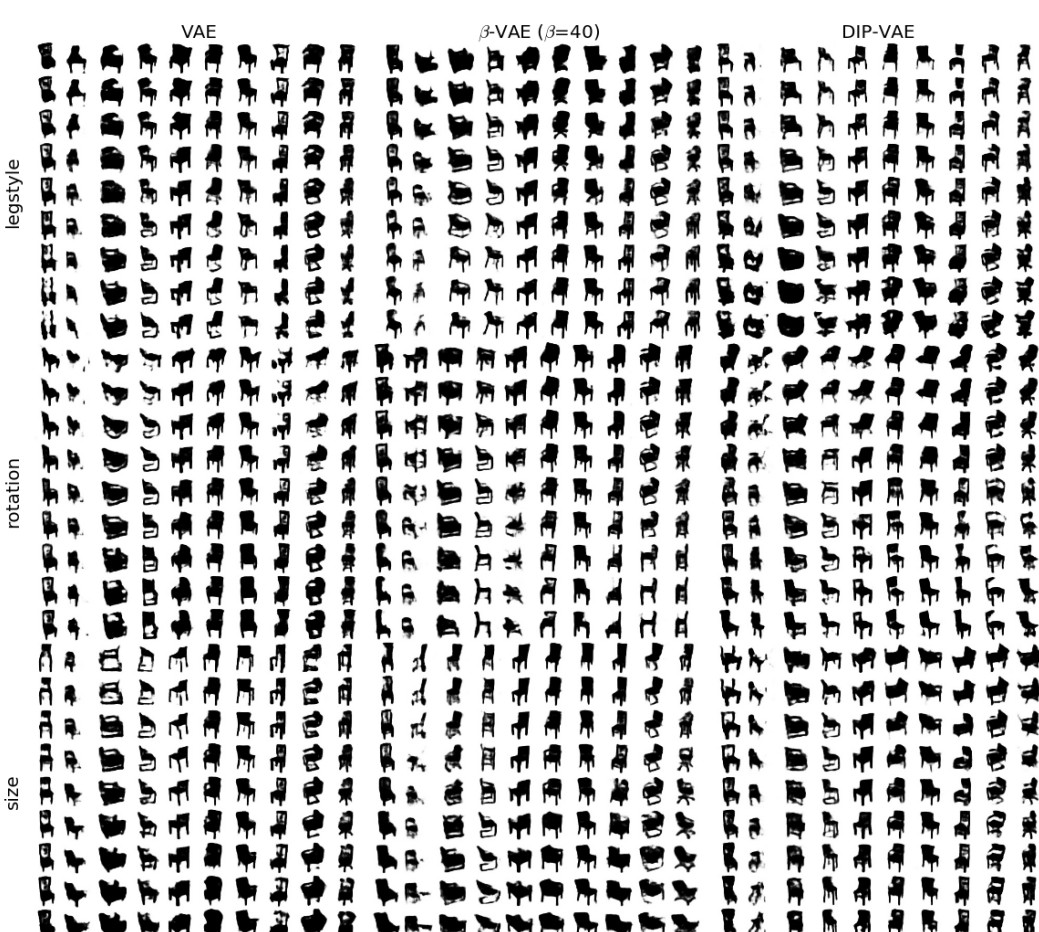

Figure 6: Qualitative results for disentanglement in Chairs dataset.

