# OpenReview forum: "Variational Inference of Disentangled Latent Concepts from Unlabeled Observations"
_ICLR.cc/2018/Conference — Accept (Poster)_

### Official Review · AnonReviewer2 · 2017-11-15
**Potentially good paper, but the presentation of the results needs improvements**

**Rating:** 7
**Confidence:** 4

**Review:**

This paper describes DIP-VAE, an improvement on the beta-VAE framework for learning a disentangled representation of the data generative factors in the visual domain. The authors propose to augment the standard VAE ELBO objective with an extra term that minimises the covariance between the latents. Unlike the original beta-VAE objective which implicitly minimises such covariance individually for each observation x, the DIP-VAE objective does so while marginalising over x. This difference removes the tradeoff between reconstruction quality and disentangling reported for beta-VAE, since DIP-VAE maintains sensitivity of q(z|x) to each observation x, and hence achieves disentangling while preserving the sharpness of reconstructions.

Pros:
- the paper is well written
- it makes a contribution to an important line or research (unsupervised disentangled representation learning)
- the covariance minimisation proposed in the paper looks like an easy to implement yet impactful change to the VAE objective to encourage disentanglement while preserving reconstruction quality
- it directly compares the performance of DIP-VAE to that of beta-VAE showing significant improvements in terms of disentangling metric and reconstruction error

Cons:
- I am yet to be fully convinced how well the approach works. Table 1 and Figure 1 look good, but other figures are either tangental to the main point of the paper, or impossible to read due to the small scale. For example, the qualitative evaluation of the latent traversals is almost impossible due to the tiny scale of Table 5 (shouldn't this be a Figure rather than a Table?)
- The authors concentrate a lot on the CelebA dataset, however I believe the comparison with beta-VAE would be a lot clearer on the dSprites dataset (https://github.com/deepmind/dsprites-dataset) (the authors call it 2D shapes). I would like to see latent traversals of the best DIP-VAE vs beta-VAE to demonstrate good disentangling and the improvements in reconstruction quality. This (and larger Table 5) would be better use of space compared to Table 2 and Figure 2 for example, which I feel are somewhat tangental to the main message of the paper and are better suited for the appendix.
- I wonder how the authors calculated the disentanglement metric on CelebA, given that the ground truth attributes in the dataset are often rather qualitative (e.g. attractiveness), noisy (many can be considered an inaccurate description of the image), and often do not align with the data generative factors discoverable through unsupervised modeling of the data distribution
- Table 3 - the legends for the axes are too small and are impossible to read. Also it would be helpful to normalise the scales of the heat plots in the second row.
- Table 3 -  looking at the correlations with the ground truth factors, it seems like beta-VAE did not actually disentangle the latents. Would be nice to see the corresponding latent traversal plots to ensure that the baseline is actually trained well.

I am willing to increase my score for the paper if the authors can address my points. In particular I would like to see a clear comparison in terms of latent traversals on dSprites between beta-VAE and DIP-VAE models presented in Table 3. I would also like to see where these particular models lie in Figure 1.

---------------------------
---- UPDATE ----------
---------------------------
I have increased my score after reading the revised version of the manuscript.

---

> ### Author Response · Authors · 2018-01-01
> **Author response**
>
> Thank you for the careful reading of the paper and thoughtful comments! Your comments encouraged us to look carefully into the alignment of Disentanglement metric score of [Higgins et al, 2017] (which we refer to as “Z-diff score” in the revised version) with the quality of latent traversal plots. We found that it is not a reliable indicator of disentanglement, more so in the case of 2D Shapes (dsprites) data, and we were so far using the Z-diff score to pick the best model. We have taken this time to revise the paper as follows:
>
> [[Latent traversal plots, along with a new metric for measuring disentanglement -- SAP Score:]]
> We added a small section (Sec 3) describing a novel metric for measuring disentanglement, referred as Separated Attribute Predictability (SAP) score. It works by fitting a slope/intercept (for regression) or a threshold (real number, for classification) to predict each of the known generative factors (or attributes) using each latent dimension individually. This gives us a matrix S of size (# latent-dims x # attributes) indicating goodness of linear fit of the generative factors to the individual latent dimensions. For each attribute (column of S) we take the difference of top two scores and average these for all attributes to get the final SAP score. We observe that this score is a much better indicator of disentanglement seen in the decoder’s output for the single latent traversals. It is also easier to compute and does not involve training any classifier. We present the plots of both Z-diff score and SAP score vs the reconstruction error in the revised version (Fig. 1 and 2), along with the latent traversal plots for 2D Shapes data (Fig. 3 and 4).
> It is clear from Fig 3 (and from Fig 1 SAP scores vs reconstruction) that VAE is the worst in terms of disentanglement. \beta-VAE (beta=4) improves over it for disentanglement but DIP-VAE-II (lambda=5) outperforms it with better disentanglement and reconstruction quality. \beta-VAE (beta=60) provides even better disentanglement (Fig 3) but scale remains entangled with X-pos and Y-pos. DIP-VAE-II (lambda=500) yields less reconstruction error and better disentanglement in latent traversals with less entangling b/w scale and Y-pos/ X-pos.
>
> [[Two variants of DIP-VAE:]]
> We have added another variant (DIP-VAE-II, Eq. 7) in the revised version that yields better SAP-score / reconstruction-error tradeoff (and latent traversals) for 2D Shapes data (please see Fig 1). Note that in terms of Z-diff score / reconstruction error trade-off, DIP-VAE-I is still better than DIP-VAE-II for 2D Shapes data, which led us to conclude that Z-diff score is not a good metric for disentanglement.
>
> [[Disentanglement metric (Z-diff score) for CelebA:]]
> We compute this score for CelebA in the same manner as for 2D Shapes. Although some of the CelebA attributes cannot be taken as “generative factors”, we still observe that disentanglement metric scores are correlated well with the disentanglement seen in the latent traversal plots.
>
> [[Correlation plots in the submitted version:]]
> The label-correlation plots shown in the submitted version for \beta-VAE (beta=4 and 20) were for the fully-trained models after convergence (as can be seen in Fig 3, the generated images from the decoder are good). For these \beta values, the latent dimensions are indeed entangled as also reflected in the SAP score (Fig 1 in the revised version) and in the latent traversal plots (Fig 3 in the revised version). However, we later found out that in the submitted version, \beta-VAE for \beta=60, was not trained to convergence. We have fixed it in the revised version and \beta=60 gives the best SAP and Z-diff scores for \beta-VAE along with good disentanglement in the latent traversals (Fig 3), although with bad reconstruction quality. We have omitted correlation plots in the revised version as SAP score already captures it as the explained variance of linear regression fit.

---

> > ### Comment · AnonReviewer2 · 2018-01-02
> > **Good revision**
> >
> > Thank you for submitting the revised version of your manuscript. I am happy to increase my score given the new SAP metric and the improved plots.
> >
> > Minor final comments:
> >
> > 1) It would be good to include a short discussion at the end of the newly added SAP metric section that talks about how this metric might apply to 'distributed disentangled representations', where a single latent factor might be encoded by an independent set of latents units (e.g. if latent units z_{1-3} encode position x, units z_{4-6} encode position y etc).
> >
> > 2) It might be good to update the reference to the dSprites dataset to the one suggested here: https://github.com/deepmind/dsprites-dataset
> >
> > Thank you!

---

> > > ### Author Response · Authors · 2018-01-03
> > > **Updated the SAP score section**
> > >
> > > Thank you for taking time to look at the revised version! We added the following text at the end of Sec 3 discussing the scenario you mentioned:
> > > "Further, a low SAP score does not rule out good disentanglement in cases when two (or more) latent dimensions might be correlated strongly with the same generative factor and poorly with other generative factors. The generated examples using single latent traversals may not be realistic for such models, and DIP-VAE discourages this from happening by enforcing decorrelation of the latents. However, the SAP score computation can be adapted to such cases by  grouping the latent dimensions based on correlations and getting the score matrix at group level, which can be fed as input to the second step to get the final SAP score. "
> > >
> > > We have also updated the dSprites reference. Thanks!

---

### Official Review · AnonReviewer1 · 2017-11-27
**Interesting idea, some concerns on the empirical studies**

**Rating:** 7
**Confidence:** 4

**Review:**

########## UPDATED AFTER AUTHOR RESPONSE ##########

Thanks for the good revision and response that addressed most of my concerns. I am bumping up my score.

###############################################


This paper presents a Disentangled Inferred Prior (DIP-VAE) method for learning disentangled features from unlabeled observations following the VAE framework. The basic idea of DIP-VAE is to enforce the aggregated posterior q(z) = E_x [q(z | x)] to be close to an identity matrix as implied by the commonly chosen standard normal prior p(z). The authors propose to moment-match q(z) given it is hard to minimize the KL-divergence between q(z) and p(z). This leads to one additional term to the regular VAE objective (in two parts, on- and off-diagonal). It has the similar property as beta-VAE (Higgins et al. 2017) but without sacrificing the reconstruction quality. Empirically the authors demonstrate that DIP-VAE can effectively learn disentangled features, perform comparably better than beta-VAE and at the same time retain the reconstruction quality close to regular VAE (beta-VAE with beta = 1).

The paper is overall well-written with minor issues (listed below). I think the idea of enforcing an aggregated (marginalized) posterior q(z) to be close to the standard normal prior p(z) makes sense, as opposed to enforcing each individual posterior q(z|x) to be close to p(z) as (beta-)VAE objective suggests. I would like to make some connection to some work on understanding VAE objective (Hoffman & Johnson 2016, ELBO surgery: yet another way to carve up the variational evidence lower bound) where they derived something along the same line of an aggregated posterior q(z). In Hoffman & Johnson, it is shown that KL(q(z) | p(z)) is in fact buried in ELBO, and the inequality gap in Eq (3) is basically a mutual information term between z and n (the index of the data point). Similar observations have led to the development of VAMP-prior (Tomczak & Welling 2017, VAE with a VampPrior). Following the derivation in Hoffman & Johnson, DIP-VAE is basically adding a regularization parameter to the KL(q(z) | p(z)) term in standard ELBO. I think this interpretation is complementary to (and in my opinion, more clear than) the one that’s described in the paper.

My concerns are mostly regarding the empirical studies:

1. One of my main concern is on the empirical results in Table 1. The disentanglement metric score for beta-VAE is suspiciously lower than what’s reported in Higgins et al., where they reported a 99.23% disentanglement metric score on 2D shape dataset. I understand the linear classier is different, but still the difference is too large to ignore. Hence my current more neutral review rating.

2. Regarding the correlational plots (the bottom row of Table 3 and 4), I don’t think I can see any clear patterns (especially on CelebA). I wonder what’s the point of including them here and if there is a point, please explain them clearly in the paper.

3. Figure 2 is also a little confusing to me. If I understand the procedure correctly, a good disentangled feature would imply smaller correlations to other features (i.e., the numbers in Figure 2 should be smaller for better disentangled features). However, looking at Figure 2 and many other plots in the appendix, I don’t think DIP-VAE has a clear win here. Is my understanding correct? If so, what exactly are you trying to convey in Figure 2?

Minor comments:

1. In Eq (6) I think there are typos in terms of the definition of Cov_q(z)(z)? It appears as only the second term in Eq (5).

2. Hyperparameter subsection in section 3: Shouldn’t \lambda_od be larger if the entanglement is mainly reflected in the off-diagonal entries? Why the opposite?

3. Can you elaborate on how a running estimate of Cov_p(x)(\mu(x)) is maintained (following Eq (6)). It’s not very clear at the current state of the paper.

4. Can we have error bars in Table 2? Some of the numbers are possibly hitting the error floor.

5. Table 5 and 6 are not very necessary, unless there is a clear point.

---

> ### Author Response · Authors · 2018-01-01
> **Author response**
>
> Thank you for the careful reading of the paper and thoughtful comments! We were not aware of the work [Hoffman & Johnson 2016, ELBO Surgery] and it looks like the method can also be motivated from that perspective. However we are not sure about your note “the inequality gap in Eq (3) is basically a mutual information term between z and n (the index of the data point)” -- it looks like the gap between KL(q(z|x)||p(z)) and KL(q(z)||p(z)) is the mutual information term between z and n, whereas the Inequality (3) compares KL(q(z|x)||p(z|x)) and KL(q(z)||p(z)).
>
> [[Disentanglement metric score on \beta-VAE:]]
> We found out that \beta-VAE for \beta=60 was not trained to convergence which now gives the best metric score for \beta-VAE on 2DShapes data (95.7%). We have updated the Table 1 and Figure 1 with new results. [Higgins et al, 2017] report 99.23% score on 2D shape which is close to what we get. Apart from the linear classifier, the difference could also be due to the evaluation protocol where in [Higgins et al, 2017]  that trained 30 \beta-VAE models with different random seeds and “discarded the bottom 50% of the thirty resulting scores and reported the remaining results” (quoting verbatim from [Higgins et al, 2017]). We also discovered in this duration that the metric proposed in [Higgins et al, 2017] is not a good indicator of disentanglement seen in the latent traversal plots (ie, decoder’s output by varying one latent while fixing others). We added a short section (Sec 3) on the new metric we propose (referred as Separated Attribute Predictability or SAP score) which is much better aligned with the subjective disentanglement we see in the latent traversals. We have also added plots for SAP score vs reconstruction error (Fig 1 and 2).
>
> [[Correlation plots:]]
> We agree that these plots were not conveying any insights or quantitative measure for disentanglement. We have omitted them in the revised version.
>
> [[Fig 2 in the submitted version:]]
> As CelebA dataset has many ground truth attributes which are correlated with each other, it is not possible to infer different dimensions of latents capturing these (at least with the current approaches). Through this plot we were trying to show that the top attributes corresponding to a given dimension are semantically more similar for our method compared to the baselines. As you rightly noticed this is a subjective question so we have omitted these plots in the revised version.
>
> [[Cov_{q(z)}[z] in Eq 6:]]
> The first term in Cov_{q(z)}[z] in Eq 5 is a diagonal matrix (expectation of variance of variational posterior, which is a Gaussian with diagonal covariance) and contributes only to the variances of z~q(z), so in the regularizer we had considered only the second term Cov_{p(x)} [\mu(x)] which is a dense square matrix. However we have now included another variant (DIP-VAE-II) where the regularizer uses complete Cov_{q(z)}[z]. This actually provides better results on 2D Shapes data.
>
> [[Hyperparameters \lambda_od and \lambda_d:]]
> We have included a discussion on this in the paragraph after Eq 5. Essentially, penalizing the off-diagonal entries of Cov_{p(x)} [\mu(x)] also ends up reducing the diagonals of this matrix as off-diagonal are really derived from the diagonals (product of square root of diagonals for each example followed by averaging over examples). Hence holding the diagonals to a fixed value was important. We found that \lambda_d > \lamda_od was better for decreasing the covariance without impacting the variance.
>
>
>
>
> [[Running estimate of Cov_p(x)(\mu(x)):]]
> If the estimate using the current minibatch is B and the previous cumulative estimate is C, we take a combination B + a*C with ‘a’ being the inertia parameter (0.95 or so) and then normalize by (1/1-a). C is treated as constant while backpropagating the gradients.

---

> > ### Comment · AnonReviewer1 · 2018-01-09
> > **Thanks for the response and revision**
> >
> > Following the response and revision, I am bumping up the score.
> >
> > Regarding my comment about the "inequality gap in Eq (3)", I indeed meant the gap between KL(q(z|x) || p(z)) and KL(q(z) || p(z)), not Eq (3).

---

> > > ### Author Response · Authors · 2018-01-10
> > > **Thanks**
> > >
> > > Thanks for taking time to look at the revised version!

---

### Official Review · AnonReviewer3 · 2017-11-27
**The authors propose an interesting twist on regularizers for decorrelating the posterior dimensions of VAE-type models.**

**Rating:** 6
**Confidence:** 5

**Review:**

******
Update: revising reviewer score to 6 after acknowledging revisions and improved manuscript
******

The authors propose a new regularization term modifying the VAE (Kingma et al 2013) objective to encourage learning disentangling representations.
Specifically, the authors suggest to add penalization to ELBO in the form of -KL(q(z)||p(z)) , which encourages a more global criterion than the local ELBOs.
In practice, the authors decide that the objective they want to optimize is unwieldy and resort to moment matching of covariances of q(z) and p(z) via gradient descent.
The final objective uses a persistent estimate of the covariance matrix of q and upgrades it at each mini-batch to perform learning.

The authors use this objective function to perform experiments measuring disentanglement and find minor benefits compared to other objectives in quantitative terms.

Comments:
1. The originally proposed modification in Equation (4) appears to be rigorous and as far as I can tell still poses a lower bound to log(p(x)). The proof could use the result posed earlier: KL(q(z)||p(z)) is smaller than E_x KL(q(z|x)||p(z|x)).
2. The proposed moment matching scheme performing decorrelation resembles approaches for variational PCA and especially independent component analysis. The relationship to these techniques is not discussed adequately. In addition, this paper could really benefit from an empirical figure of the marginal statistics of z under the different regularizers in order to establish what type of structure is being imposed here and what it results in.
3. The resulting regularizer with the decorrelation terms could be studied as a modeling choice. In the probabilistic sense, regularizers can be seen as structural and prior assumptions on variables. As it stands, it is unnecessarily vague which assumptions this extra regularizer is making on variables.
4. Why is using the objective in Equation (4) not tried and tested and compared to? It could be thought that subsampling would be enough to evaluate this extra KL term without any need for additional variational parameters \psi. The reason for switching to the moment matching scheme seems not well motivated here without showing explicitly that Eq (4) has problems.
5. The model seems to be making on minor progress in its stated goal, disentanglement. It would be more convincing to clarify the structural properties of this regularizer in a statistical sense more clearly given that experimentally it seems to only have a minor effect.
6. Is there a relationship to NICE (Laurent Dinh et al)?
7. The infogan is also an obvious point of reference and comparison here.
8. The authors claim that there are no models which can combine GANs with inference in a satisfactory way, which is obviously not accurate nowadays given the progress on literature combining GANs and variational inference.

All in all I find this paper interesting but would hope that a more careful technical justification and derivation of the model would be presented given that it seems to not be an empirically overwhelming change.

---

> ### Author Response · Authors · 2018-01-01
> **Author response**
>
> Thank you for the careful reading of the paper and thoughtful comments!
>
> [[Lower bound to log p(x):]]
> The proposed objective is indeed a lower bound to the evidence log p(x). However the proof is really trivial and doesn’t need to use Inequality (3). It can simply be shown by using nonnegativity of the distance between q(z) and p(z) (KL or any other divergence) -- since standard ELBO is a lower bound, subtracting any nonnegative quantity is also a lower bound.
>
> [[Optimizing objective (4):]]
> Optimizing (4) with KL(q(z)||p(z)) ( = E_{q(z)} log (q(z)/p(z))) is not tractable with sampling because of the \log q(z) term (q(z)=E_{p(x)} q(z|x)). One possibility (as we discuss in the paper) is to use variational form of KL as first proposed in (Nguyen et al., 2010) and later in (Nowozin et al., 2016) for GANs, however it will involve training a separate “discriminator” that is expected to approximate the KL divergence at every iteration, which can be minimized using backprop. We have tried this recently and found that the covariance matrix of z~q(z) has significant off-diagonal entries (ie, it does not decorrelate the latents well).
>
> [[Structural properties / assumptions in the regularizer:]]
> The proposed regularizer KL(q(z)||p(z)) is trying to match the inferred prior q(z) to the hypothesized prior p(z), which will automatically happen if p_\theta(x) is close to p(x) (the model is good) and q(z|x) is close to p(z|x). We have also discussed this in the paragraph just after Eq (3). The regularizer is trying to enforce this explicitly which is totally natural (unless the hypothesized prior itself is too unreasonable).
>
> [[Relationship to NICE (Dinh et al, 2016):]]
> NICE is another framework to do density estimation with latent variables where encoder and decoder are exact inverses of each other by design (encoder maps from data distribution to a factored distribution of same dimensionality following a specific architecture that allows for easy inverse computation, and hence easy sampling as well as tractable maximum likelihood based learning). It can be compared/contrasted with the VAE which our proposed method is based upon -- eg, the reconstruction error is zero by design. We restrict ourselves to the VAE in this work.
>
> [[Comparison with InfoGAN:]]
> Unlike VAE, InfoGAN does not have a trained inference mechanism for real observations. Though it has a network Q (shared with discriminator) that implicitly minimizes E_{x~G(z)} KL(p(z|x) || q(z||x)), this is targeted towards inference for fake examples from the generator. [Higgins et al, 2017] compare \beta-VAE with InfoGAN finding that \beta-VAE outperforms and one of the reasons could be the lack of a true inference model.
>
> [[Inference in GANs:]]
> We are aware of ALI [Dumoulin et al, ‎2017] and BiGAN [Donahue et al, 2017] which still suffer from bad reconstruction (D(E(x)) is far from x) as observed in [Kumar et al, 2017]. There is also a recent work [Arora et al, 2017] showing that the encoder in ALI/BiGAN may potentially learn non-informative codes.
>
> [Arora et al, 2017] Theoretical limitations of Encoder-Decoder GAN architectures, arXiv:1711.02651 2017
> [Dumoulin et al, ‎2017] Adversarially learned inference, ICLR 2017
> [Donahue et al, 2017] Adversarial feature learning, ICLR 2017
> [Kumar et al, 2017] Semi-supervised Learning with GANs: Manifold Invariance with Improved Inference, NIPS 2017

---

### Public Comment · (anonymous) · 2017-11-24
**Where does the \Sigma_phi(x) term go in the moment-matching penalty? Comparisons to InfoGAN?**

i'm wondering why you have omitted the \Sigma_phi(x) term in equation (6). Is this a typo?

Moreover, isn't it worth comparing to InfoGAN (Chen et al, 2016), which seems to be competitive against beta-VAE at least for cases where you have both discrete and continuous factors of variation? As far as I'm aware, InfoGANs are designed to work with joint continuous and discrete latents, whereas beta-VAE only uses continuous latents so less well-suited for capturing discrete factors of variation. You seem to use binary attributes in the celebA dataset for computing the measure of disentanglement (bangs/male/beard/hat).

---

> ### Author Response · Authors · 2018-01-02
> **response**
>
> Please take a look at the revised version for Eq 6 and another variant.
>
> InfoGAN does not have a trained inference mechanism for real observations. Though it has a network Q (shared with discriminator) that implicitly minimizes E_{x~G(z)} KL(p(z|x) || q(z||x)), this is targeted towards inference for fake examples from the generator. [Higgins et al, 2017] compare \beta-VAE with InfoGAN finding that \beta-VAE outperforms it, and one of the reasons could be the lack of a true inference model. However, your point regarding continuous vs discrete latents is valid to some extent as reparameterization trick doesn't work with discrete variables. One can use Gumbel-Softmax / Concrete distribution to approximate a discrete distribution for which reparameterization can be used but we don't explore this in our current work.

---

### Decision · Program_Chairs · 2018-01-29
**ICLR 2018 Conference Acceptance Decision**

**Decision:**

Accept (Poster)

**Comment:**

Thank you for submitting you paper to ICLR. The reviewers are all in agreement that the paper is suitable for publication, each revising their score upwards in response to the revision that has made the paper stronger.

The authors may want to consider adding a discussion about whether the simple standard Gaussian prior, which is invariant under transformation by an orthogonal matrix, is a sensible one if the objective is to find disentangled representations. Alternatives, such as sparse priors, might be more sensible if a model-based solution to this problem is sought.